# Heart-Focused Anxiety, General Anxiety, Depression and Health-Related Quality of Life in Patients with Atrial Fibrillation Undergoing Pulmonary Vein Isolation

**DOI:** 10.3390/jcm11071751

**Published:** 2022-03-22

**Authors:** Valérie Pavlicek, Sonja Maria Wedegärtner, Dominic Millenaar, Jan Wintrich, Michael Böhm, Ingrid Kindermann, Christian Ukena

**Affiliations:** Klinik für Innere Medizin III, Kardiologie, Angiologie und Internistische Intensivmedizin, Universitätsklinikum des Saarlandes, D-66421 Homburg, Germany; sonja.wedegaertner@gmail.com (S.M.W.); dominic.millenaar@uks.eu (D.M.); jan.wintrich@uks.eu (J.W.); michael.boehm@uks.eu (M.B.); ingrid.kindermann@uks.eu (I.K.); christian.ukena@uks.eu (C.U.)

**Keywords:** atrial fibrillation, pulmonary vein isolation, depression, anxiety, quality of life

## Abstract

(1) Background: Atrial fibrillation (AF) is associated with anxiety, depression, and chronic stress, and vice versa. The purpose of this study was to evaluate potential effects of pulmonary vein isolation (PVI) on psychological factors. (2) Methods: Psychological assessment was performed before PVI as well as after six months. (3) Results: A total of 118 patients [age 64 ± 9 years, 69% male, left ventricular ejection fraction 57 ± 8%, 56% paroxysmal AF] undergoing PVI were included. After PVI, significant improvements were observed in the mean total heart-focused anxiety (HFA) score, as well as in the Cardiac Anxiety Questionnaire (CAQ) sub-scores: HFA attention, HFA fear, and HFA avoidance scores. Subgroup analyses showed an association of improvement with freedom of documented AF recurrence. Mean scores of general anxiety and depression evaluated by the Hospital Anxiety and Depression Scale (HADS) decreased significantly after PVI in all subgroups regardless of AF recurrence. Further, both physical and mental composite scores of the Short Form Health Survey (SF-12) increased significantly from baseline. (4) Conclusions: PVI results in a significant reduction in HFA. Improvements in general anxiety and depressive symptoms did not seem to be related only to rhythm control per se. Therefore, CAQ may represent a more specific evaluation tool as HADS in patients with AF.

## 1. Introduction

Atrial fibrillation (AF) is the most prevalent arrhythmia worldwide affecting more than 33 million people with a global incidence of approximately 5 million newly diagnosed AF patients each year [1]. A further increase is anticipated in the coming decades [2]. The global burden-of-disease study showed a twofold increase in AF-related mortality from 1990 to 2010 [1]. AF is associated with a higher relative risk of all-cause mortality, stroke, cardiovascular mortality, cardiac events, and heart failure, and a number of risk factors were described [3,4]. There is increasing evidence indicating a close interaction between psychological factors and AF disease condition. A few reviews analyzed the link between anxiety and depression suggesting a condition of comorbidity [5,6,7]. Since the pulmonary veins have been identified as triggers of atrial fibrillation, pulmonary vein isolation (PVI) has become the mainstream of rhythm control treatment strategies. Current guidelines recommend a catheter ablation AF after one failed therapy with a class I or III antiarrhythmic drug to improve symptoms in patients with paroxysmal or persistent AF as well as in patients with decreased LV-function due to tachycardia-induced cardiomyopathy independent of their symptom status (class I indication). As a first-line rhythm control therapy, PVI should be considered to improve symptoms in selected patients with symptomatic paroxysmal AF (IIa) and persistent AF without major risk factors for AF recurrence (IIb) as an alternative to antiarrhythmic drugs class I or III, considering patient choice, benefit, and periprocedural risk [8]. Only a few studies showed that catheter ablation results in a sustainable reduction in psychological distress and an improvement of quality of life [9,10]. Further, data on heart-focused anxiety (HFA) exist regarding patients with chronic heart-failure and coronary artery disease but not for patients with AF [11,12,13]. Therefore, this study aimed to determine how treatment with AF catheter ablation leads to changes in HFA, general anxiety, and depression, as well as health-related quality of life (HRQoL). Further we performed the study to search for a valuable tool to identify AF patients with anxiety and depression, because the presence of these psychological disorders may impact the effectiveness of AF treatment. Identifying AF patients with psychological comorbidities as anxiety and depression may improve AF treatment outcomes by implementing strategies reducing anxiety and depression.

## 2. Materials and Methods

Study Design: A total of 389 consecutive patients with paroxysmal or persistent AF undergoing PVI (radiofrequency (RF) or cryoablation) at Saarland University center were screened between December 2016 and April 2019. Inclusion criteria were the presence of symptomatic AF; first PVI procedure; age of 18 years or older; informed consent. Patients undergoing Re-PVI procedure (*n* = 56), incomplete psychological assessment, or lacking informed consent were excluded from the analysis (*n* = 215). Finally, a total of 118 patients were included in our monocentric, prospective clinical study (Figure 1). Symptoms, psychosocial, and demographic factors were assessed by standardized and validated psychological methods before scheduled PVI as well as six months after the procedure, e.g., heart-focused anxiety (HFA) by the Cardiac Anxiety Questionnaire (CAQ), general anxiety and depression by the Hospital Anxiety and Depression Scale (HADS), and HRQoL by the Short Form Health Survey (SF-12). The study was approved by the local institutional review board. Informed consent to participate in this study, was obtained in all cases. The study is based on ethical guidelines of the Declaration of Helsinki.

Pre-procedural work-up: All patients underwent 12-lead-ECG, transthoracic echocardiography (TTE). Transesophageal echocardiography or contrast enhanced computed tomography were performed to confirm the absence of intracardiac thrombi and review left atrial anatomy before the PVI procedure.

Ablation procedure: (a) Cryoablation: With an over-the-wire technique with a circular mapping catheter (Achieve mapping catheter, Medtronic Inc., Dublin, Ireland), the 28 mm diameter second-generation cryoballoon (Medtronic Inc.) was used for antral ablation of the pulmonary veins. A minimum of one freeze lasting between 180 to 240 s was performed at the ostium of each pulmonary vein, with isolation confirmed by entrance and/or exit block with the circular mapping catheter. (b) Radiofrequency (RF) ablation: RF ablation was performed with guidance from a CARTO 3-dimensional mapping system (Biosense Webster, Irvine, CA, USA) using a double Lasso technique. A radiofrequency current was delivered by a ThermoCool catheter (Biosense Webster) with a power maximum of 30 W. Endpoint was a bidirectional conduction block between the left atrium and the pulmonary vein.

Measures: The type and intensity of AF-related symptoms were measured with a non-standardized questionnaire. This questionnaire contained 5 items (palpitations, shortness of breath, dizziness, sleepiness, and chest pain) with a five-point Likert-type scale answer categories (0 = no, 1 = mild, 2 = moderate, 3 = severe, 4 = very severe symptoms).

The Cardiac Anxiety Questionnaire (CAQ) is a 17-item self-report inventory scored in a five-point Likert-type scale, ranging from 0 (never) to 4 (always) [14]. Higher scores indicate greater anxiety. The CAQ allows us to measure the overall characteristic value of HFA and three sub-scores: heart-focused fear, attention, and avoidance with age-dependent and sex-dependent cut-off scores. Cut-off scores are provided in Appendix A. We chose the ninetieth percentile as a cut-off for clinically significant HFA [15].

The Hospital Anxiety and Depression Scale (HADS) is a frequently used 14-item self-rating scale developed to assess psychological distress in non-psychiatric patients. It contains two subscales: anxiety (7 items) and depression (7 items). Each item is coded from 0 to 3. The scores for anxiety and depression can vary from 0 to 21, depending on the severity of the symptoms. Zigmond et al. have proposed the following thresholds which were used in this study: a score from 0 to 7 does not indicate the presence or symptoms of anxiety or depression; a score from 8 to 10 indicates the presence of moderate symptoms (therefore doubtful cases); a score ≥ 11 indicates clinically significant symptoms of anxiety or depression [16,17].

A 12-Item Short Form Health Survey (SF-12) was used to assess HRQoL. The SF-12 consists of two scores: a physical and a psychological/mental impairment score [18]. A higher score indicates a better health status [19]. The estimated values were compared to values of the German general population [20].

Statistical analyses: Data are presented as mean ± standard deviation (SD), median (interquartile range), or number (percentage) unless otherwise specified. Baseline comparisons of groups (men vs. women, median age <64 vs. ≥64, no AF recurrence vs. AF recurrence at 6 months follow-up) were performed using the Pearson Chi-square test for categorical variables and the Wilcoxon rank sum test, the Kruskal–Wallis H test, or the *t*-test for continuous variables. For the baseline to follow-up within group comparisons, paired tests were appropriate. For the analysis of interactions between time (baseline vs. follow-up) and group repeated measures, ANOVA was performed. A two-tailed *p* value of <0.05 was regarded as statistically significant. All statistical analyses were performed with SPSS statistical software (version 25.0, SPSS Inc., Chicago, IL, USA).

## 3. Results

### 3.1. Baseline Characteristics

Baseline patients’ characteristics before PVI (baseline) are summarized in Table 1. Psychological measures as well as single comparisons of subgroups are provided in Table 2. The mean age of the 118 included patients was 64 ± 9 years (69% male). Fifty-five patients (47%) underwent cryoablation, and sixty-three patients (53%) underwent RF ablation of the pulmonary veins (*p* = 0.49). All patients were symptomatic (60% EHRA II, 37% EHRA III and 3% EHRA IV; *p* = 0.37). Sixty-six patients (56%) suffered from paroxysmal AF, and fifty-two patients (44%) from persistent AF (*p* = 0.23) with a mean CHA_2_DS_2_-VAsc-Score of 2.9 ± 1.7. Fifty-one patients (43%) were on therapy with antiarrhythmic drugs (AAD). Mean left ventricular ejection fraction (LVEF) was 57 ± 8%. Compared to men (*n* = 81), women (*n* = 37) were older (68 ± 8 years vs. 61 ± 10 years; *p* < 0.01), had a higher CHA_2_DS_2_-Vasc-Score (3.9 ± 1.3 vs. 2.4 ± 1.7; *p* < 0.01) and suffered from hypertension more often (89% vs. 63%; *p* < 0.01). Patients with persistent AF had a lower LVEF (55 ± 8% vs. 58 ± 8%, *p* = 0.03) and had higher rates of hypertension (84% vs. 69%; *p* = 0.02) and diabetes (27% vs. 9%; *p* = 0.01) compared to patients with paroxysmal AF.

### 3.2. Type and Intensity of Symptoms before and after Pulmonary Vein Isolation

Overall, PVI led to a significant symptom reduction (palpitations, shortness of breath, dizziness, sleepiness, and chest pain) on a five-point Likert-type scale (0 = no, 1 = mild, 2 = moderate, 3 = severe, 4 = very severe symptoms) despite patients having documented AF recurrence (Table 3).

### 3.3. Psychological Status and Quality of Life before and after Pulmonary Vein Isolation

Mean baseline scores (standard deviation) of the psychological variables, as well as single comparison of subgroups, are provided in Table 2. Results of the baseline to 6 months follow-up comparison of HFA total and related sub-scores for all patients and subgroups are presented in Figure 2. Figure 3 shows the results for the comparison of general anxiety, depression, and HRQoL scores between baseline and 6 months follow-up for the whole sample as well as subgroups.

#### 3.3.1. Heart-Focused Anxiety and Related Subscores

Before PVI, 45% of participants had clinically relevant HFA (total score). HFA-related attention, fear, and avoidance were increased in 40%, 40%, and 22% of the patients. Significant improvements after PVI were observed in the mean total HFA score [1.71 ± 0.61 at baseline vs. 1.33 ± 0.63 at 6 months follow-up; *p* < 0.01], as well as in the HFA attention [1.92 ± 0.71 at baseline vs. 1.48 ± 0.64 at 6 months follow-up; *p* < 0.01], fear [1.78 ± 0.70 at baseline vs. 1.42 ± 0.77 at 6 months follow-up; *p* < 0.01], and avoidance score [1.34 ± 1.04 at baseline vs. 0.95 ± 0.94 at 6 months follow-up; *p* < 0.01]. Six months after PVI, 25% of patients reported clinically relevant HFA (total score). HFA attention, fear, and avoidance were increased in 18%, 22%, and 9% of the participants (Figure 2).

#### 3.3.2. Subgroup Analyses for Heart-Focused Anxiety Outcomes

At baseline, female participants reported slightly higher significant HFA (total score) and HFA avoidance than males [1.89 ± 0.64 vs. 1.63 ± 0.59, *p* = 0.04 and 1.63 ± 0.59 vs. 1.20 ± 0.99, *p* = 0.04]. Furthermore, in older patients (≥64 years) and patients with persistent AF, significantly higher values for HFA avoidance were observed [1.63 ± 1.05 vs. 1.03 ± 0.94, *p* < 0.01 and 1.75 ± 1.06 vs. 1.01 ± 0.90, *p* < 0.01] compared to younger patients (<64 years) and patients with paroxysmal AF, respectively (Table 2).

After PVI, total HFA, HFA attention, HFA fear, and HFA avoidance significantly improved in all subgroups except in patients with documented AF recurrence. Patients without a documented recurrence of AF reported a significantly stronger improvement in HFA attention compared to patients with AF reccurence at 6 months follow-up (Figure 2).

#### 3.3.3. General Anxiety, Depression, and Health-Related Quality of Life

At baseline, increased levels of general anxiety were reported by 63% of the patients. Further, 23% of the participants suffered from clinically relevant anxiety symptoms. Depressive symptoms were increased in 52% with clinical relevance in 24% of patients. Mean scores of general anxiety and depression decreased significantly after PVI [8.76 ± 3.45 at baseline vs. 7.08 ± 3.06 at 6 months follow-up; *p* < 0.01 and 7.95 ± 3.49 at baseline vs. 6.75 ± 3.90 at 6 months follow-up; *p* < 0.01]. Six months after PVI, in 39% of patients, increased levels for general anxiety were observed, and 16% of the patients reported clinically relevant anxiety symptoms. The depressive symptoms also decreased after PVI (depressive symptoms in 30%, and clinically relevant depressive symptoms in 16% of patients). From the index ablation to the 6 months follow-up visit, the composite physical health scores significantly increased by 4 points [40 to 44 points, respectively; *p* < 0.01] as well as the composite mental health scores by 3 points [46 to 49 points, respectively; *p* = 0.03] (Table 2).

#### 3.3.4. Subgroup Analyses for General Anxiety, Depression, and Health-Related Quality of Life

At baseline, the subgroups did not differ significantly in mean general anxiety and depression scores. In older patients (≥64 years) and patients with persistent AF, significantly lower physical health scores were observed [37 ± 11 vs. 44 ± 10, *p* < 0.01 and 38 ± 10 vs. 42 ± 12, *p* = 0.04] (Table 2).

After PVI, mean general anxiety and depression scores improved in all subgroups. Only the mean depression score in young patients (<64 years) did not change significantly. Young patients (<64 years) and patients with no documented AF recurrence showed an improvement in the physical composite score [44 vs. 49 (*p* < 0.01), 40 vs. 44 (*p* < 0.01) and 39 vs. 46 (*p* < 0.01)]. Further, the subgroup analyses showed no significant improvement in terms of the mental composite score (Figure 3).

## 4. Discussion

In our monocentric, prospective clinical study, we analyzed the effects of catheter ablation on heart-focused and general anxiety, depression, and health-related quality of life in patients with paroxysmal and persistent AF. Overall, this study shows that HFA as a psychological dimension appears to be as relevant for individuals with cardiac disease as it has been shown for certain groups without heart disease.

So far, data on HFA exist regarding patients with chronic heart-failure and coronary artery disease but not for patients with AF [11,12,13]. Therefore, the evaluation of heart-focused anxiety in patients with AF represents new data. We were able to confirm our assumption that HFA decreases after PVI. Before PVI, 45% of participants had clinically relevant HFA (total score). HFA-related attention, fear, and avoidance were increased in 40%, 40%, and 22% of the patients, respectively. In comparison, for patients with chronic heart failure before defibrillator implantation, as previously described by Kindermann et al., clinically relevant HFA (total score) was reported by 44%, and HFA-related attention, fear, and avoidance by 38%, 26%, and 44% of the patients before device implantation, respectively [12]. Six months after PVI, still 25% of the patients reported clinically relevant HFA (total score). HFA attention, fear, and avoidance were increased in 18%, 22%, and 9% of the participants, respectively. Identifying this subgroup of patients is important, so they could be offered psychotherapeutic treatment or other supportive treatment to adjust more successfully to their cardiac disease, and HFA should be included in the behavioral management of AF patients.

The patients’ general psychological condition (general anxiety, depression, and HRQoL) improved after catheter ablation. At baseline, increased levels of general anxiety were reported by 63% of the patients. Further, 23% of the participants suffered from clinically relevant anxiety symptoms. Depressive symptoms were increased in 52% with clinical relevance in 24% of patients. In comparison to AF patients before catheter ablation, a smaller number of increased levels of general anxiety (37%) and increased depressive symptoms (33%) were reported by patients with chronic heart failure before defibrillator implantation, respectively [12]. Six months after catheter ablation, in 39% of patients, increased levels of general anxiety were observed and 16% of the patients reported clinically relevant anxiety symptoms. The depressive symptoms also decreased after PVI (depressive symptoms in 30%, and clinically relevant depressive symptoms in 16% of patients). From the index ablation to the six months follow-up visit, the composite physical health scores significantly increased by 4 points as well as the composite mental health scores by 3 points. The STOP-AF Post-Approval Study also showed an increase by 5 points in the physical component and of 4 points in the mental component of SF-12 six months after cryoablation [10].

As depression was associated with the recurrence of AF after PVI [21], this result may be of particular importance. In terms of general anxiety no association with AF recurrence or ablation was reported before [22]. Neither do our results indicate an association with general anxiety. The increased levels of general anxiety can be partly attributed to the upcoming PVI, because the HADS only assess the severity of anxiety symptoms during the last seven days. Interestingly, referring to heart-focused anxiety, our data revealed no improvements in patients with recurrent AF six months after PVI possibly due to HFA representing a specific pattern of anxiety comprising the fear of cardiac-related sensations [14]. This applied to the total score as for any sub-score as fear, attention, and avoidance. They further did not benefit in terms of physical quality of life. On the other hand, improvements were observed in all other subgroups and the overall sample. A current study points to the significant correlation between AF and heart-focused anxiety [11]. These results underline the need of greater emphasis on heart-focused anxiety in patients with AF. The change of general anxiety might be attributed to a sham-like effect of the intervention per se. However, only HFA was capable of reflecting psychological changes following an improvement of AF symptoms.

The ABC pathway as a novel management strategy intergrating all aspects of AF treatment has been promoted to improve outcomes in patients with AF. Besides stroke prevention and reduction in cardiovascular risk factors, it includes education and life-style changes [23]. Identifying anxiety and/or depression as described above as well as implementing strategies which can reduce anxiety and depression in AF patients may improve treatment outcomes, patient quality of life, and possibly reduce financial burden to the health care system associated with AF. Patient education of the disease process, reduction in uncertainty, and perhaps treating patients with psychiatric medications such as antidepressants could represent such strategies. However, the data on using antidepressants in AF patients to control anxiety and depression symptoms or to prevent AF is very limited. Medical treatment of anxiety and depression in AF patients may also be challenging due to possible interactions between antidepressant, antiarrhythmic drugs, and anticoagulants. There are many unanswered questions on how to manage AF patients with anxiety and/or depression. Further trials are needed to elucidate the benefits of patient education or behavioral therapy as well as medical treatment of psychological disorders in the prevention of incident AF, and the impact in terms of reduction in anxiety and depression levels in AF patients.

The primary indication for AF ablation is better symptom control. We observed a significant reduction in all estimated AF-related symptoms at follow-up. The effect on symptom improvement following PVI is in line with other studies [24]. A recent meta-analysis showed that first-line treatment with AF ablation is superior to antiarrhythmic drugs therapy in patients with symptomatic paroxysmal atrial fibrillation. AF ablation significantly reduced the recurrence of any symptomatic atrial arrhythmias and healthcare resource utilization with comparable safety profile [25].

In addition to a better symptom control, the concept of AF ablation as first-line treatment for AF is strongly related to quality of life which may also play an important role in all psychological aspects in the future [26].

However, our study has certain limitations. The estimate of residual AF burden after PVI was based on frequent Holter and in intermittent 12-lead electrocardiograms and not by using implantable monitoring, which possibly underestimated of the post ablation AF recurrence. Nonetheless, this approach to post-interventional rhythm monitoring is common clinical practice. Further, the follow-up time of six months only provides data in intermediate-term outcome of catheter ablation and much higher AF recurrence rates were previously described. Of the ablation patients in the CABANA trial, for example, 12.6% had a recurrence of symptomatic AF and 36.4% had experienced a recurrence of any AF by 12 months [27]. Another limitation is the use of generic HRQoL questionnaires (SF-12) and the measurement of AF-related symptoms by a non-standardized questionnaire. These tools have limited resolution to determine symptoms related to AF [28]. The majority of patients received beta-blockers, but the potential effects of these on anxiety and depression were not considered in our study. Furthermore, the interpretation of the results is limited due to the lack of a control group, the unknown “time-to-diagnosis” of AF as well as the “time-to-PVI”, and the relatively small size of the overall study group, thus reducing generalizability.

## 5. Conclusions

PVI for treatment of symptomatic AF results in a significant reduction in heart-focused anxiety, general anxiety, and depressive symptoms, as well as in an improvement in HRQoL. The improvement may be mediated by a reduction in symptoms and AF burden. As depressive symptoms and anxiety are common in patients with atrial fibrillation, healthcare providers should monitor patients with AF for depressive symptoms and anxiety at the time of catheter ablation and intervene when indicated. The assessment of HFA by the CAQ may represent a more specific evaluation tool than HADS in patients with AF. Further investigations in the assessment, prediction, treatment, and outcome of depressive symptoms and anxiety are needed in the future to ensure optimal care for patients with AF.

## Figures and Tables

**Figure 1 jcm-11-01751-f001:**
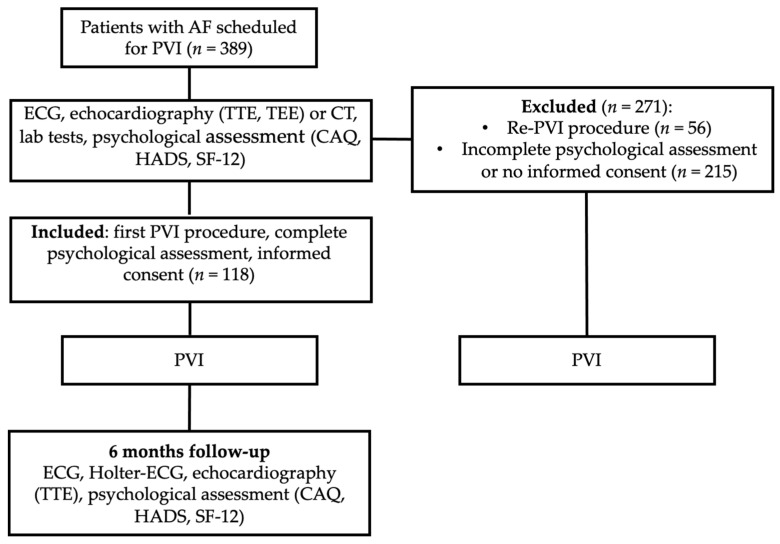
Study Flow Chart. AF = atrial fibrillation; PVI = pulmonary vein isolation; ECG = electrocardiogram; TTE = transthoracic echocardiography; TEE = transesophageal echocardiography, CT = computed tomography; CAQ = Cardiac Anxiety Questionnaire; HADS = Hospital Anxiety and Depression Scale; SF-12 = Short Form Health Survey.

**Figure 2 jcm-11-01751-f002:**
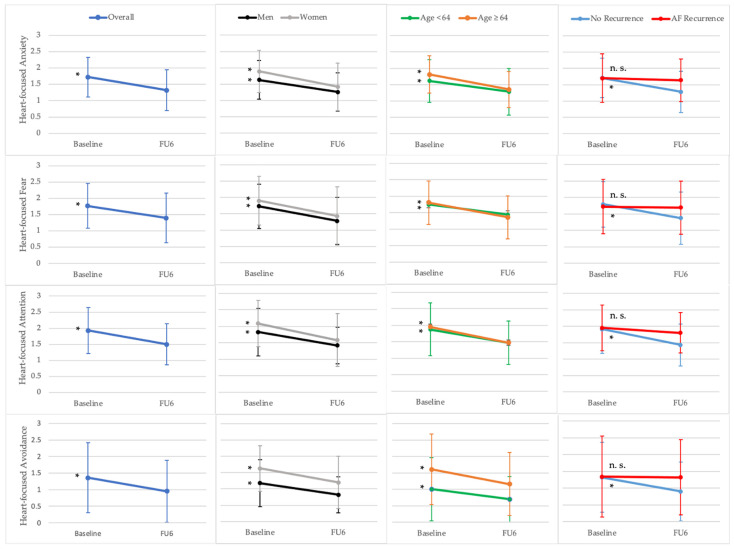
Heart-focused anxiety (HFA) with related sub-scores at baseline and 6 months follow-up (FU6). Results for paired *t*-tests and repeated measures ANOVA with the within-subject factor time (baseline, FU6) and the between-subject factor group (gender, age, documented atrial fibrillation (AF) recurrence). * *p* < 0.05 for paired *t*-test; n.s., not significant. Data points represent mean scores of the Cardiac Anxiety Questionnaire (CAQ) and sub-scores (HFA anxiety (total score), HFA fear, HFA attention and HFA avoidance). The error bars indicate the standard deviation.

**Figure 3 jcm-11-01751-f003:**
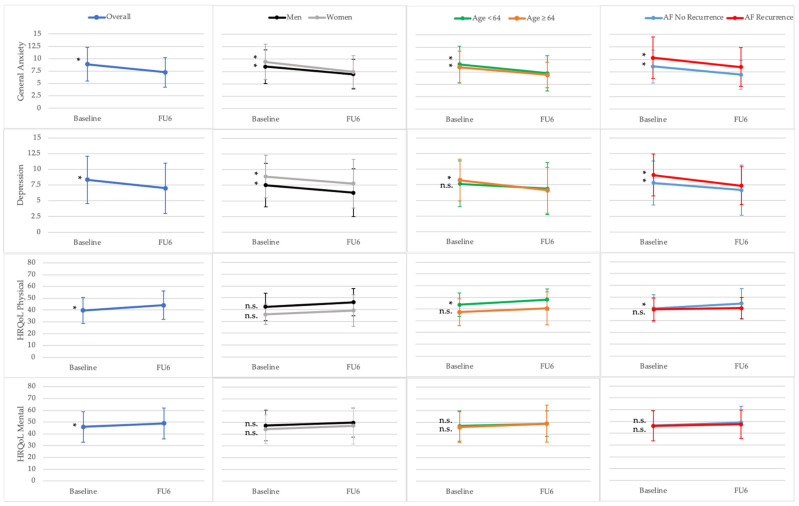
General anxiety, depression, and Health-related Quality of life (HRQoL) at baseline and 6 months follow-up (FU6). Results for paired *t*-tests and repeated measures ANOVA with the within-subject factor time (baseline, FU6) and the between-subject factor group (gender, age, documented atrial fibrillation (AF) recurrence). * *p* < 0.05 for paired *t*-test; n.s., not significant. Data points represent mean scores of the Hospital Anxiety and Depression Scale (General Anxiety and Depression) and of the 12-Item Short Form Health Survey. The error bars indicate the standard deviation.

**Table 1 jcm-11-01751-t001:** Baseline characteristics.

	Total	Sex		*p*	Age		*p*	AF Recurrence > 3 Months Post PVI		*p*
		Female	Male		<64 Years	≥64 Years		No Recurrence	Recurrence	
*n* (%)	118	37 (31)	81 (69)	<0.01	57 (48)	61 (52)	0.78	105 (89)	13 (11)	<0.01
Age (years), mean ± SD	64 ± 9	68 ± 8	61 ± 10	<0.01	55 ± 6	71 ± 5	<0.01	63 ± 10	65 ± 11	0.55
Ablation modality, *n* (%)				0.49			0.87			
Cryoablation	55 (47%)	19 (51)	36 (44)		27 (47)	28 (46)		49 (47)	6 (46)	
RF ablation	63 (53%)	18 (49)	45 (56)		30 (52)	33 (54)		56 (53)	7 (54)	
LVEF (%), mean ± SD	57 ± 8	58 ± 7	56 ± 8	0.39	56 ± 9	57 ± 6	0.64	57 ± 8	57 ± 7	0.99
EHRA class, *n* (%)				0.37			0.09			0.38
II	71 (60)	21 (57)	50 (62)		30 (53)	41 (67)		65 (62)	6 (46)	
III	44 (37)	16 (43)	28 (34)		24 (42)	20 (33)		37 (35)	7 (54)	
IV	3 (3)	0	3 (4)		3 (5)	0		3 (3)	0	
CHA_2_DS_2_-Vasc-Score, mean ± SD	2.9 ± 1.7	3.9 ± 1.3	2.4 ± 1.7	<0.01	1.8 ± 1.4	3.9 ± 1.4	<0.01	2.8 ± 1.8	3.2 ± 1.9	0.48
CVRF, *n* (%)										
Hypertension	85 (72)	33 (89)	52 (64)	<0.01	34 (69)	51 (84)	<0.01	74 (71)	11 (85)	0.28
Diabetes mellitus	20 (17)	6 (16)	14 (17)	0.89	7 (12)	13 (21)	0.19	16 (15)	4 (31)	0.16
Dyslipidaemia	66 (56)	23 (62	43 (53)	0.36	27 (47)	39 (64)	0.07	56 (53)	10 (77)	0.11
Coronary artery disease, *n* (%)	27 (23)	10 (27)	17 (21)	0.47	11 (19)	16 (26)	0.37	25 (24)	2 (15)	0.50
CKD (Crea-GFR < 60 mL/min/1.73 m^3^)	26 (22)	11 (30)	15 (19)	0.17	5 (9)	21 (34)	<0.01	22 (21)	4 (31)	0.42
Prior stroke or TIA, *n* (%)	12 (10)	3 (8)	9 (11)	0.62	5 (9)	7 (12)	0.63	12 (11)	0	0.20
Sleep apnoea, *n* (%)	11 (9)	1 (3)	10 (12)	0.09	7(12)	4 (7)	0.29	11 (11)	0	0.22
BMI (kg/m^2^), mean ± SD	29 ± 5	28 ± 5	29 ± 4	0.24	29 ± 5	28 ± 5	0.17	29 ± 5	27 ± 4	0.22
Medication, *n* (%)										
Beta-blocker	104 (88)	34 (92)	70 (86)	0.39	46 (81)	58 (95)	0.02	94 (90)	10 (77)	0.19
Antiarrhythmic	51 (43)	15 (41)	36 (44)	0.69	24 (42)	27 (44)	0.81	47 (45)	4 (31)	0.34
ACE inhibitor	36 (31)	12 (32)	24 (30)	0.76	18 (32)	18 (30)	0.81	33 (31)	3 (23)	0.54
AT_1_ antagonist	46 (39)	15 (41)	31 (38)	0.82	20 (35)	26 (43)	0.40	39 (37)	7 (54)	0.24
Aldosterone antagonist	13 (11)	5 (14)	8 (10)	0.52	7 (12)	6 (10)	0.70	13 (12)	0	0.18

AF, atrial fibrillation; SD, standard deviation; PVI, pulmonary vein isolation; RF, radiofrequency; LVEF, left ventricular ejection fraction; EHRA, European Heart Rhythm Association; CVRF, cardiovascular risk factors; CKD, chronic kidney disease; TIA, transient ischemic attack; BMI, body mass index; ACE, angiotensin-converting enzyme; AT, angiotensin.

**Table 2 jcm-11-01751-t002:** Psychological measures at baseline and 6 months follow-up.

	Total	Sex		*p*	Age		*p*	AF Recurrence > 3 Months Post PVI		*p*
		Female	Male		<64 Years	≥64 Years		No Recurrence	Recurrence	
*n* (%)	118	37 (31)	81 (69)	<0.01	57 (48)	61(52)	0.78	105 (89)	13 (11)	<0.01
Cardiac Anxiety Questionnaire (CAQ)										
Baseline										
HFA anxiety, mean ± SD	1.71 ± 0.61	1.89 ± 0.64	1.63 ± 0.59	0.04	1.61 ± 0.65	1.82 ± 0.56	0.07	1.72 ± 0.60	1.70 ± 0.75	0.92
HFA attention, mean ± SD	1.92 ± 0.71	2.10 ± 0.69	1.83 ± 0.71	0.06	1.87 ± 0.78	1.96 ± 0.64	0.52	1.91 ± 0.72	1.95 ± 0.69	0.88
HFA fear, mean ± SD	1.78 ± 0.70	1.88 ± 0.73	1.73 ± 0.69	0.29	1.73 ± 0.76	1.82 ± 0.65	0.54	1.78 ± 0.69	1.72 ± 0.60	0.77
HFA avoidance, mean ± SD	1.34 ± 1.04	1.63 ± 0.59	1.20 ± 0.99	0.04	1.03 ± 0.94	1.63 ± 1.05	<0.01	1.34 ± 1.03	1.34 ± 1.21	0.99
6 months follow-up										
HFA anxiety, mean ± SD	1.33 ± 0.63	1.44 ± 0.73	1.28 ± 0.58	0.21	1.29 ± 0.72	1.37 ± 0.54	0.51	1.29 ± 0.63	1.65 ± 0.60	0.05
HFA attention, mean ± SD	1.48 ± 0.64	1.61 ± 0.81	1.43 ± 0.55	0.15	1.47 ± 0.66	1.49 ± 0.64	0.88	1.45 ± 0.65	1.74 ± 0.59	0.13
HFA fear, mean ± SD	1.42 ± 0.77	1.44 ± 0.89	1.41 ± 0.72	0.85	1.46 ± 0.90	1.38 ± 0.64	0.60	1.38 ± 0.76	1.75 ± 0.79	0.11
HFA avoidance, mean ± SD	0.95 ± 0.94	1.22 ± 0.98	0.83 ± 0.89	0.04	0.70 ± 0.86	1.17 ± 0.96	<0.01	0.90 ± 0.91	1.33 ± 1.06	0.12
Hospital Anxiety and Depression Scale (HADS)										
Baseline										
Anxiety, mean ± SD	8.76 ± 3.45	9.43 ± 3.57	8.36 ± 3.37	0.15	9.05 ± 3.70	8.49 ± 3.20	0.38	8.57 ± 3.32	10.31 ± 4.19	0.09
Depression, mean ± SD	7.95 ± 3.49	8.89 ± 3.41	7.52 ± 3.46	0.05	7.65 ± 3.66	8.23 ± 3.32	0.37	7.81 ± 3.49	9.08 ± 3.35	0.22
6 moths follow-up										
Anxiety, mean ± SD	7.08 ± 3.06	7.41 ± 3.25	6.94 ± 2.98	0.44	7.25 ± 3.52	6.93 ± 2.58	0.58	6.91 ± 2.92	8.46 ± 3.87	0.09
Depression, mean ± SD	6.75 ± 3.90	7.76 ± 3.90	6.28 ± 3.83	0.06	6.89 ± 4.18	6.61 ± 3.64	0.69	6.67 ± 3.40	7.38 ± 3.02	0.53
12-Item Short Form Health Survey (SF-12)										
SF-12 physical component, mean ± SD	40 ± 11	36 ± 9	43 ± 12	<0.01	44 ± 10	37 ± 11	< 0.01	40 ± 12	40 ± 9	0.78
SF-12 mental component, mean ± SD	46 ± 13	43 ± 12	47 ± 13	0.16	46 ± 13	46 ± 13	0.74	46 ± 13	46 ± 13	0.95
6 months follow-up										
SF-12 physical component, mean ± SD	44 ± 13	39 ± 14	46 ± 12	<0.01	48 ± 9	40 ± 14	<0.01	44 ± 13	41 ± 9	0.29
SF-12 mental component, mean ± SD	49 ± 13	47 ± 16	50 ± 12	0.34	49 ± 11	49 ± 16	0.97	49 ± 14	48 ± 12	0.70

PVI, pulmonary vein isolation; AF, atrial fibrillation; SD, standard deviation; HFA, heart-focused anxiety.

**Table 3 jcm-11-01751-t003:** Type and intensity of symptoms at baseline and 6 months follow-up.

	Total	Sex		*p*	Age		*p*	AF		*p*	AF Recurrence > 3 Months Post PVI		*p*
		Female	Male		<64 Years	≥64 Years		Paroxysmal	Persistent		No Recurrence	Recurrence	
*n* (%)	118	37 (31)	81 (69)	<0.01	57 (48)	61 (52)	0.78	66 (56)	52 (44)	0.23	105 (89)	13 (11)	<0.01
Palpitations, mean ± SD													
Baseline	3.07 ± 1.19	3.38 ± 0.95	2.93 ± 1.24	0.06	3.07 ± 1.25	3.06 ± 1.11	0.97	3.23 ± 1.12	2.85 ± 1.22	0.09	3.09 ± 1.41	2.90 ± 1.52	0.64
6 months follow-up	1.70 ± 0.90	2.00 ± 1.09	1.68 ± 0.97	0.14	1.82 ± 0.95	1.74 ± 1.09	0.69	1.71 ± 0.92	1.87 ± 1.14	0.41	1.70 ± 0.93	2.38 ± 1.45	0.02
*p* (baseline vs. 6 months follow-up)	<0.01	<0.01	<0.01		<0.01	<0.01		<0.01	<0.01		<0.01	0.10	
Shortness of breath, mean ± SD													
Baseline	2.37 ± 1.30	2.62 ± 1.44	2.30 ± 1.21	0.23	2.29 ± 1.20	2.50 ± 1.38	0.40	2.07 ± 1.22	2.81 ± 1.27	<0.01	2.39 ± 1.31	2.50 ± 1.08	0.79
6 months follow-up	1.75 ± 0.94	2.09 ± 1.10	1.68 ± 0.89	0.04	1.78 ± 0.99	1.83 ± 0.96	0.80	1.77 ± 0.91	1.85 ± 1.06	0.68	1.74 ± 0.96	2.31 ± 0.95	0.047
*p* (baseline vs. 6 months follow-up)	<0.01	0.02	<0.01		<0.01	<0.01		0.02	<0.01		<0,01	0.52	
Chest pain, mean ± SD													
Baseline	1.85 ± 0.98	1.94 ± 0.89	1.82 ± 1.02	0.54	1.85 ± 0.99	1.85 ± 0.97	1.00	1.79 ± 0.99	1.94 ± 0.95	0.44	1.86 ± 0.98	1.80 ± 1.03	0.86
6 months follow-up	1.43 ± 0.79	1.37 ± 0.77	1.54 ± 0.85	0.32	1.53 ± 0.90	1.45 ± 0.75	0.61	1.42 ± 0.81	1.57 ± 0.85	0.34	1.45 ± 0.82	1.77 ± 0.83	0.19
*p* (baseline vs. 6 months follow-up)	< 0.01	<0.01	0.01		<0.01	<0.01		<0.01	0.02		<0.01	1.00	
Dizziness, mean ± SD													
Baseline	2.00 ± 1.02	2.15 ± 1.11	1.97 ± 1.02	0.42	2.07 ± 1.12	1.98 ± 0.97	0.65	1.89 ± 0.94	2.21 ± 1.15	0.11	2.03 ± 1.06	2.00 ± 0.94	0.93
6 months follow-up	1.78 ± 0.91	1.86 ± 0.94	1.85 ± 0.94	0.95	1.87 ± 0.90	1.83 ± 0.98	0.80	1.88 ± 0.87	1.81 ± 1.04	0.70	1.83 ± 0.94	2.00 ± 0.91	0.54
*p* (baseline vs. 6 months follow-up)	0.02	0.09	0.08		0.05	0.13		0.68	<0.01		0.02	0.51	
Sleepiness, mean ± SD													
Baseline	2.30 ± 1.21	2.24 ± 1.23	2.35 ± 1.23	0.66	2.40 ± 1.13	2.22 ± 1.31	0.45	2.11 ± 1.12	2.57 ± 1.32	0.05	2.27 ± 1.21	2.70 ± 1.34	0.29
6 months follow-up	1.90 ± 0.99	1.89 ± 1.13	1.95 ± 0.99	0.77	2.13 ± 1.11	1.74 ± 0.93	0.047	1.86 ± 0.99	2.02 ± 1.09	0.43	1.88 ± 1.00	2.31 ± 1.18	0.16
*p* (baseline vs. 6 months follow-up)	<0.01	0.03	<0.01		0.02	<0.01		0.03	<0.01		<0.01	0.34	

SD, standard deviation.

## Data Availability

The data used to support the findings of this study are included within the article.

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
