# Peer review of "Heart-Focused Anxiety, General Anxiety, Depression and Health-Related Quality of Life in Patients with Atrial Fibrillation Undergoing Pulmonary Vein Isolation"

_jcm, 2022, doi:10.3390/jcm11071751_

Round 1
Reviewer 1 Report
The author has revised the manuscript, I am happy with the current version. Thank you for your hard work.
Author Response
Dear reviewer,
thank you for reviewing our manuscript and your approval for publication.
Sincerely, yours
V. Pavlicek
Reviewer 2 Report
1. Material and methods- paroxysmal and permanent AF(AF that is accepted by the patient and physician, and no further attempts to restore/maintain sinus rhythm will be undertaken). I assume the authors meant “persistent”
2. How the recurrence of AF was ascertained
3. Taking into account the possible difference in the perception of symptoms, and thus the level of anxiety, should we consider the division into patients with paroxysmal and persistent AF? Especially since the assessment of symptoms was based on a non-standardized questionnaire.
4. Have the authors considered the potential effects of beta-blockers on anxiety and depression?
Author Response
Please see the attachment.

This manuscript is a resubmission of an earlier submission. The following is a list of the peer review reports and author responses from that submission.
Round 1
Reviewer 1 Report
This is an interesting paper describing the changes, prominently of the psychological and psychiatric changes, but also of physical symptoms that occur following pulmonary vein isolation ablation of AF. By necessity it is open and uncontrolled as a controlled trial would not be practical. The authors explain the a number of psychological problems, and in particular heart based anxiety are frequent in patients with AF and may be an effect or contributing factor to their AF.
The majority of people reading this paper would know what the baseline score and extent of change mean in terms of clinical significance. The authors state that have compared their results with that population results for Germans. I can't see where this has been explained. It is therefore important that the authors explain in clinical terms the relative disability of the baseline scores and how much each improvement is likely to mean to the patient.
Other points:
- The clinical characteristics of the population are very descriptive of HFpEF. It would be of value to know the size of their left atria as this is also a very common indicator of HFpEF and a strong predictor of likelihood of AF recurrence.
- The failure to show psychological improvements in patients who have a recurrence of AF may be simply because the number of patients is small.
- In figure 2 the error bars should be SD not SEM. The data are demonstrating the variability on 2 occasions of measurements done in a selected group. They do not represent an attempt to calculate the closeness of the mean of a sample taken from a population to the true mean of that population.
Author Response
Dear reviewer,
Thank you for reading the manuscript and the very valuable and useful suggestions for improvement. In the following I tried to answer your questions and your suggestions were inserted into the manuscript. Unfortunately, it was not possible to format the page number in the template. I would like to apologize for that.
Kind regards.
V. Pavlicek
The majority of people reading this paper would know what the baseline score and extent of change mean in terms of clinical significance. The authors state that have compared their results with that population results for Germans. I can't see where this has been explained. It is therefore important that the authors explain in clinical terms the relative disability of the baseline scores and how much each improvement is likely to mean to the patient.
The Cardiac Anxiety Questionnaire served to measure HFA, including 17 items rated on a 5-point Likert-scale from 0 (never) to 4 (always).1 It yields a total HFA score and three subscores (HFA-fear, HFA-attention, and HFA-avoidance) with age-dependent and sex-dependent cut-off scores2.
1Eifert G.H., Zvolensky M.J., Lejuez C.W. Heart-Focused Anxiety and Chest Pain: A Conceptual and
Clinical Review. Clinical Psychology: Science and Practice. 2000; 7: 403-417. doi: 10.1093/clipsy.7.4.403.
2 Fischer D., Kindermann I., Karbach J., Herzberg P.Y., Ukena C., Barth C., Lenski M., Mahfoud F., Einsle F., Dannemann S., Böhm M., Köllner V. Heart-focused anxiety in general population. Clin. Res. Cardiol. 2012; 101 (2): 109-16. doi: 10.1007/s00392-011-0371-7.
The manuscript was changed on page 2, line 89ff as follows:
“Cardiac Anxiety Questionnaire (CAQ) is a 17-item self-report inventory scored in a five-point Likert-type scale, ranging from 0 (never) to 4 (always) [11]. The CAQ allows measuring the overall characteristic value of HFA and three sub-scores heart-focused (HFA) fear, attention, and avoidance. Higher scores indicate greater anxiety. Levels on each scale were compared to the German general population. We chose the ninetieth percentile as cut-off for clinically significant HFA [12].”
to
“Cardiac Anxiety Questionnaire (CAQ) is a 17-item self-report inventory scored in a five-point Likert-type scale, ranging from 0 (never) to 4 (always) [11]. Higher scores indicate greater anxiety. The CAQ allows measuring the overall characteristic value of HFA and three sub-scores heart-focused (HFA) fear, attention, and avoidance with age-dependent and sex dependent cut-off scores. Cut-off scores are provided in Table 1 of the supplementary material. We chose the ninetieth percentile as cut-off for clinically significant HFA [12].”
A table with the HFA cut-off values has also been added in the supplementary material on page 14.
Supplementary material
Table 1. Cut-off scores for the Cardiax Anxiety Questionnaire (CAQ)
|
|
|
|
Age dependent |
|||
|
|
|
|
18-43 |
44-54 |
55-66 |
67-92 |
|
CAQ |
0-4 |
|
|
|
|
|
|
Total (male) |
|
|
0.94 |
1.41 |
1.70 |
1.92 |
|
Total (female) |
|
|
1.21 |
1.35 |
1.82 |
2.00 |
|
Fear (male) |
|
|
1.25 |
1.50 |
1.88 |
1.96 |
|
Fear (female) |
|
|
1.50 |
1.50 |
2.00 |
2.13 |
|
Attention (male) |
|
|
1.00 |
1.60 |
2.00 |
2.10 |
|
Attention (female) |
|
|
1.20 |
1.40 |
1.82 |
2.20 |
|
Avoidance (male) |
|
|
1.25 |
1.50 |
2.00 |
2.75 |
|
Avoidance (female) |
|
|
1.50 |
2.00 |
2.00 |
2.75 |
|
CAQ: Cardiac Anxiety Questionnaire. Cut-off scores refer to Fischer et al. for CAQ11. |
||||||
Other points:
- The clinical characteristics of the population are very descriptive of HFpEF. It would be of value to know the size of their left atria as this is also a very common indicator of HFpEF and a strong predictor of likelihood of AF recurrence.
That´s a very good point, but the atrial size as predictor of AF recurrence was not the focus of the study.
2. The failure to show psychological improvements in patients who have a recurrence of AF may be simply because the number of patients is small.
That is absolutely correct. It was mentioned on page 12 in the discussion part as one of the limitations of the study.
“Furthermore, the interpretation of the results is limited due to the lack of a control group and the relatively small size of the overall study group, thus reducing generalizability.”
3. In figure 2 the error bars should be SD not SEM. The data are demonstrating the variability on 2 occasions of measurements done in a selected group. They do not represent an attempt to calculate the closeness of the mean of a sample taken from a population to the true mean of that population.
Figure 2 on page 7 and figure 3 on page 8 were replaced by new figures with error bars representing SD as follows:
Figure 2. Heart-focused anxiety (HFA) with related sub-scores at baseline and 6 months follow-up (FU6). Results for paired t-tests and repeated measures ANOVA with the within-subject factor time (baseline, 6FU) and the between-subject factor group (gender, age, documented atrial fibrillation (AF) recurrence). *p<0.05 for paired t-test; n.s., not significant. Data points represent mean scores of the Cardiac Anxiety Questionnaire (CAQ) and sub-scores (HFA anxiety (total score), HFA fear, HFA attention and HFA avoidance). The error bars indicate the standard deviation.
Figure 3. General anxiety, depression, and Health-related Quality of life (HRQoL) at baseline and 6 months follow-up (FU6). Results for paired t-tests and repeated measures ANOVA with the within-subject factor time (baseline, FU6) and the between-subject factor group (gender, age, documented atrial fibrillation (AF) recurrence). *p<0.05 for paired t-test; n.s., not significant. Data points represent mean scores of the Hospital Anxiety and Depression Scale (General Anxiety and Depression) and of the 12-Item Short Form Health Survey. The error bars indicate the standard deviation.

Reviewer 2 Report
I’m honored to review the original article entitled “Heart-focused anxiety, general anxiety, depression and health-2 related quality of life in patients with atrial fibrillation under-3 going pulmonary vein isolation” by Valérie Pavlicek et al.
They investigated how treatment with AF catheter ablation leads to changes in heart-focused anxiety (HFA), general anxiety, and depression, as well as health-related quality of life (HRQoL). They concluded that PVI for treatment of symptomatic AF results in a significant reduction of heart-fo-cused anxiety, general anxiety, and depressive symptoms as well as to an improvement in HRQoL.
I have the following comments.
Major comments:
In introduction section, you should clearly state what is not clear at this time and why did you do this research.
Incomplete psychological assessment cases is too many. It would create a bias.
You should describe follow up method because you stratified patients by with or without recurrence. I’m concern about what quality you detected AF recurrence.
What is clinical implication of this study? You should state in discussion section.
Minor comments:
There are two HFA of abbreviation in page2 line 60 and 86. This is confusing.
You should state study design clearly in method section, although you mention discussion section. I thought it is retrospective study because many patients were excluded.
Did all patients undergo only PVI?
Author Response
Dear reviewer,
Thank you for reading the manuscript and the very valuable and useful suggestions for improvement. In the following I tried to answer your questions and your suggestions were inserted into the manuscript. Unfortunately, it was not possible to format the page number in the template. I would like to apologize for that.
Kind regards.
Pavlicek
Major comments:
In introduction section, you should clearly state what is not clear at this time and why did you do this research.
Thank you for this valuable comment. The aim of the study was not completely stated as you mentioned above. Of course, showing an effect of PVI in psychological factors was not the only aim of the study. We also performed the study to search for a valuable tool to identify AF patients with anxiety and depression, because the presence of anxiety and depression may impact the effectiveness of AF treatment. Identifying AF patients with psychological comorbidities as anxiety and depression may improve AF treatment outcomes by implementing strategies reducing anxiety and depression.
This further aim of out study was added to the introduction part on page 2, line 50ff as follows:
…Further we performed the study to search for a valuable tool to identify AF patients with anxiety and depression, because the presence of these psychological disorders may impact the effectiveness of AF treatment. Identifying AF patients with psychological comorbidities as anxiety and depression may improve AF treatment outcomes by implementing strategies reducing anxiety and depression.
Incomplete psychological assessment cases is too many. It would create a bias.
You are absolutely right. In fact, more than 200 of eligible patients failed to complete their questionnaires or did not give informed consent to participate in the study. The n=389 refers to screened patients including patients undergoing a re-do-procedure (exclusion criteria). Nevertheless, about a half of these patients either did answer the questionnaires at baseline or at follow-up completely, and had to be excluded from the analysis. This observation possibly may create a bias, of course. We do not really know why so many patients were unable to answer the questionnaires correctly or answer these at all. We also critically reconsidered our method of data collection and would change our strategy next time.
You should describe follow up method because you stratified patients by with or without recurrence. I’m concern about what quality you detected AF recurrence.
Your concerns are not unfounded. In fact, the estimation of residual AF burden after PVI was based on frequent Holter and in intermittent 12-lead electrocardiograms and not by using implantable monitoring, which possibly underestimated of the post ablation AF recurrences, especially the asymptomatic ones.
What is clinical implication of this study? You should state in discussion section.
That is a legitimate question. The clinical implication was stated and supplemented in the discussion part. The manuscript was changed on page 11, line 277ff as follows:
” Identifying anxiety and/or depression as described above as well as implementing strategies which can reduce anxiety and depression in AF patients may improve treatment outcomes, patient quality of life and possibly reduce financial burden to the health care system associated with AF. Patient education of the disease process, reduction of uncertainty, and perhaps treating patients with psychiatric medications as antidepressants could represent such strategies. However, the data on using antidepressants in AF patients to control anxiety and depression symptoms or to prevent AF is very limited. Medical treatment of anxiety and depression in AF patients may also be challenging due to possible interactions between antidepressant, antiarrhythmic drugs, and anticoagulants. There are many unanswered questions on how to manage AF patients with anxiety and/or depression. Further trials are needed to elucidate the benefits of patient education or behavioral therapy as well as medical treatment of psychological disorders in the prevention of incident AF, and the impact in terms of reduction of anxiety and depression levels in AF patients.”
Minor comments:
There are two HFA of abbreviation in page2 line 60 and 86. This is confusing.
The manuscript was changed in page 2, line 91ff as follows:
“The CAQ allows measuring the overall characteristic value of HFA and three sub-scores heart-focused (HFA) fear, attention, and avoidance with age-dependent and sex-dependent cut-off scores.”
to
“The CAQ allows measuring the overall characteristic value of HFA and three sub-scores heart-focused fear, attention, and avoidance with age-dependent and sex-dependent cut-off scores.”
You should state study design clearly in method section, although you mention discussion section. I thought it is retrospective study because many patients were excluded.
I understand your concerns about the design, but the study design was prospective. We only had to exclude a very low number of patients at a later timepoint of the study due to lacking follow-up questionnaires.
The manuscript was changed on page 2, line 62ff as follows:
“A total of 118 patients were finally included (Figure 1).”
to
“Finally, a total of 118 patients were included in our monocentric, prospective clinical study (Figure 1).”
Did all patients undergo only PVI?
Yes. Only patients with AF undergoing first PVI procedure. Patients with additionally ablation of atrial flutter or Re-PVI-Procedure were not included into the analysis.

Reviewer 3 Report
Thank you for the hard work. Thank you for emphasizing the holistic approach to AF management.
Minor comments:
- I believe the "real" limitation of the study is unknown "time-to-diagnosis" and "time-to-PVI".
- Moreover, symptoms, including anxiety may be related to the PVI as a procedure - not AF per se. It might be interesting to compare with control group (AF patients managed with rate control).
- It might be worth to mention, the ABC pathway as a novel managment strategy; intergreting all aspects of AF treatment (Lip, G. The ABC pathway: an integrated approach to improve AF management. Nat Rev Cardiol 14, 627–628 (2017). https://doi.org/10.1038/nrcardio.2017.153)
- Furthermore, it might be worth to mention the AF ablation as a first-line treatment for AF - a "new" concept strongly related to quality of life which may play important role in all psychological aspects in future. A recent meta-analysis of 6 randomized clinical trials:
(Imberti JF, Ding WY, Kotalczyk A, Zhang J, Boriani G, Lip G, Andrade J, Gupta D. Catheter ablation as first-line treatment for paroxysmal atrial fibrillation: a systematic review and meta-analysis. Heart. 2021 Oct;107(20):1630-1636. doi: 10.1136/heartjnl-2021-319496. Epub 2021 Jul 14. PMID: 34261737.)
And review article
( (2021) Clinical outcomes following rhythm control for atrial fibrillation: is early better?, Expert Review of Cardiovascular Therapy, 19:4, 277-287, DOI: 10.1080/14779072.2021.1902307)
Round 2
Reviewer 2 Report
The author has revised the manuscript according to the reviewer's suggestions. As a result, I feel that the manuscript has been greatly improved.
However, I think it is still unclear what we did not know before and what’s new in this study.
There are many more papers [1-6] on quality of life, anxiety and depression than what the authors have described. What have been inadequate in previous studies, and what are the new findings of this study?
- Efremidis M, Letsas KP, Lioni L, Giannopoulos G, Korantzopoulos P, Vlachos K, Dimopoulos NP, Karlis D, Bouras G, Sideris A, Deftereos S. Association of quality of life, anxiety, and depression with left atrial ablation outcomes. Pacing Clin Electrophysiol. 2014 Jun;37(6):703-11. doi: 10.1111/pace.12420. Epub 2014 May 9. PMID: 24809737.
- Bulková V, Fiala M, Havránek S, Simek J, Skňouřil L, Januška J, Spinar J, Wichterle D. Improvement in quality of life after catheter ablation for paroxysmal versus long-standing persistent atrial fibrillation: a prospective study with 3-year follow-up. J Am Heart Assoc. 2014 Jul 18;3(4):e000881. doi: 10.1161/JAHA.114.000881. PMID: 25037195; PMCID: PMC4310368.
- Yu S, Zhao Q, Wu P, Qin M, Huang H, Cui H, Huang C. Effect of anxiety and depression on the recurrence of paroxysmal atrial fibrillation after circumferential pulmonary vein ablation. J Cardiovasc Electrophysiol. 2012 Nov;23 Suppl 1:S17-23. doi: 10.1111/j.1540-8167.2012.02436.x. Epub 2012 Sep 21. PMID: 22998234.
- Barmano N, Charitakis E, Karlsson JE, Nystrom FH, Walfridsson H, Walfridsson U. Predictors of improvement in arrhythmia-specific symptoms and health-related quality of life after catheter ablation of atrial fibrillation. Clin Cardiol. 2019 Feb;42(2):247-255. doi: 10.1002/clc.23134. Epub 2018 Dec 21. PMID: 30548275; PMCID: PMC6712386.
- Jia Z, Du X, Lu S, Yang X, Chang S, Liu J, Li J, Zhou Y, Macle L, Dong J, Ma C. Effect of Mental Health Status on Arrhythmia Recurrence After Catheter Ablation of Atrial Fibrillation. Can J Cardiol. 2019 Jul;35(7):831-839. doi: 10.1016/j.cjca.2019.02.007. Epub 2019 Feb 16. PMID: 31292081.
- Samuel M, Khairy P, Champagne J, Deyell MW, Macle L, Leong-Sit P, Novak P, Badra-Verdu M, Sapp J, Tardif JC, Andrade JG. Association of Atrial Fibrillation Burden With Health-Related Quality of Life After Atrial Fibrillation Ablation: Substudy of the Cryoballoon vs Contact-Force Atrial Fibrillation Ablation (CIRCA-DOSE) Randomized Clinical Trial. JAMA Cardiol. 2021 Nov 1;6(11):1324-1328. doi: 10.1001/jamacardio.2021.3063. PMID: 34406350; PMCID: PMC8374730.
Author Response
The author has revised the manuscript according to the reviewer's suggestions. As a result, I feel thatthe manuscript has been greatly improved.
However, I think it is still unclear what we did not know before and what’s new in this study.
There are many more papers [1-6] on quality of life, anxiety and depression than what the authors have
described. What have been inadequate in previous studies, and what are the new findings of this study?
1. Efremidis M, Letsas KP, Lioni L, Giannopoulos G, Korantzopoulos P, Vlachos K, Dimopoulos
NP, Karlis D, Bouras G, Sideris A, Deftereos S. Association of quality of life, anxiety, and
depression with left atrial ablation outcomes. Pacing Clin Electrophysiol. 2014 Jun;37(6):703-11.
doi: 10.1111/pace.12420. Epub 2014 May 9. PMID: 24809737.
2. Bulková V, Fiala M, Havránek S, Simek J, Skňouřil L, Januška J, Spinar J, Wichterle D.
Improvement in quality of life after catheter ablation for paroxysmal versus long-standing
persistent atrial fibrillation: a prospective study with 3-year follow-up. J Am Heart Assoc. 2014
Jul 18;3(4):e000881. doi: 10.1161/JAHA.114.000881. PMID: 25037195; PMCID: PMC4310368.
3. Yu S, Zhao Q, Wu P, Qin M, Huang H, Cui H, Huang C. Effect of anxiety and depression on the
recurrence of paroxysmal atrial fibrillation after circumferential pulmonary vein ablation. J
Cardiovasc Electrophysiol. 2012 Nov;23 Suppl 1:S17-23. doi: 10.1111/j.1540-8167.2012.02436.x.
Epub 2012 Sep 21. PMID: 22998234.
4. Barmano N, Charitakis E, Karlsson JE, Nystrom FH, Walfridsson H, Walfridsson U. Predictors
of improvement in arrhythmia-specific symptoms and health-related quality of life after
catheter ablation of atrial fibrillation. Clin Cardiol. 2019 Feb;42(2):247-255. doi:
10.1002/clc.23134. Epub 2018 Dec 21. PMID: 30548275; PMCID: PMC6712386.
5. Jia Z, Du X, Lu S, Yang X, Chang S, Liu J, Li J, Zhou Y, Macle L, Dong J, Ma C. Effect of Mental
Health Status on Arrhythmia Recurrence After Catheter Ablation of Atrial Fibrillation. Can J
Cardiol. 2019 Jul;35(7):831-839. doi: 10.1016/j.cjca.2019.02.007. Epub 2019 Feb 16. PMID:
31292081.
6. Samuel M, Khairy P, Champagne J, Deyell MW, Macle L, Leong-Sit P, Novak P, Badra-Verdu
M, Sapp J, Tardif JC, Andrade JG. Association of Atrial Fibrillation Burden With Health-Related
Quality of Life After Atrial Fibrillation Ablation: Substudy of the Cryoballoon vs Contact-Force
Atrial Fibrillation Ablation (CIRCA-DOSE) Randomized Clinical Trial. JAMA Cardiol. 2021
Nov 1;6(11):1324-1328. doi: 10.1001/jamacardio.2021.3063. PMID: 34406350; PMCID:
PMC8374730.
Dear reviewer,
thank you for the prompt response. We understand, that there are further published studies describing
quality of life, general anxiety and depression in patients with AF undergoing PVI. In this context, our
results are in line with the previous studies. But so far, data on heart-focused anxiety (HFA) exist
regarding patients with chronic heart-failure and coronary artery disease but not for patients with AF.
Therefore, the evaluation of heart-focused anxiety in patients with AF represents new data. “We were
able to confirm our assumption that HFA decreases after PVI. Before PVI, 45% of participants had
clinically relevant HFA (total score). HFA related attention, fear, and avoidance were increased in 40%,
40%, and 22% of the patients, respectively. In comparison, in patients with chronic heart failure before
defibrillator implantation clinically relevant HFA (total score) is prevalent in44%. The sub-units HFA
related attention, fear, and avoidance was reported in 38%, 26%, and 44% of the patients before device
implantation, respectively[18]. In the present investigation, six months after PVI, still 25% of patients
reported clinically relevant HFA (total score). HFA attention, fear, and avoidance were increased in
18%, 22% and 9% of the participants, respectively. Identifying this subgroup of patients is important, as
psychotherapeutic treatment or other supportive treatment could be offered to these patients to adjust
more successfully to their cardiac disease. Therefore, assessment of HFA should be included in the
behavioral management of AF patients “(page 10, line 237ff). Meanwhile, scores of general anxiety and
depression evaluated by the Hospital Anxiety and Depression Scale (HADS) decreased significantly
after PVI in all subgroups regardless of AF recurrence. Improvements in the mean total heart-focused
anxiety (HFA) score, as well as in the Cardiac Anxiety Questionnaire (CAQ) sub-scores attention, fear
and avoidance scores showed an association of improvement with freedom of documented AF
recurrence. Therefore, the assessment of HFA by the CAQ may represent a more specific evaluation tool
compared to general anxiety and depression by the HADS in patients with AF.
These changes were made to the revised manuscript:
The following sentence has been inserted on page 2, line 48: “... Further, data on heart-focused anxiety
(HFA) exist regarding patients with chronic heart-failure and coronary artery disease but not for
patients with AF [11]–[13] ...”
The following sentences have been inserted on page 10, line 238: “...So far, data on HFA exist regarding
patients with chronic heart-failure and coronary artery disease but not for patients with AF [11]–[13].
Therefore, the evaluation of heart-focused anxiety in patients with AF represents new data ... “
References on page 12ff have been updated as follows:
References
1. Chugh S., Havmoeller R., Narayanan K, Singh D., Rienstra M., Benjamin E. Gillum R. Kim Y. McAnulty J.,
Zheng Z., Forouzanfar M., Naghavi M., Mensah G., Ezzati M., Murray C. Worldwide epidemiology of atrial
fibrillation: A global burden of disease 2010 study. Circulation. 2014; 129 (8): 837-47. doi:
10.1161/CIRCULATIONAHA.113.005119.
2. Di Carlo A., Bellino L., Consoli D., Mori F. Zaninelli A., Baldereschi M., Cattarinussi A., D ́Alfonso M.G.,
Gradia C., Sgheri B., Pracucci G., Piccardi B., Polizzi B., Inzitari D. Prevalence of atrial fibrillation in the Italian
elederly population and projections from 2020 to 2016 for Italy and the European Union: the FAI Project.
Europace. 2019; 21: 1468-1475. doi: 10.1093/europace/euz141.
3. Emdin C.A., Wong C.X., Hsiao A.J., Altman D.G., Peters S.A.E., Woodward M., Odutayo A.A. BMJ. 2016; 352:
h7013. doi: 10.1136/bmj.h7013.
4. Lau D., Nattel S., Kalman J., Sanders P. Modifiable Risk Factors and Atrial Fibrillation. Circulation. 2017; 136
(6): 583-596. doi: 10.1161/CIRCULATIONAHA.116.023163.
5. Galli F., Borghi L., Carugo S., Cavicchioli M., Faioni E.M., Negroni M.S., Vegni E. Atrial fibrillation and
psychological factors: a systematic review. PeerJ. 2017; e3537. doi: 10.7717/peerj.3537.
6. McCabe P.J. Psychological Distress in Patients Diagnosed with Atrial Fibrillation. J. Cardiovasc. Nurs. 2010; 25
(1): 40-51. doi: 10.1097/JCN.0b013e3181b7be36.
7. Patel D., Mc Conkey N.D., Sohaney R., Mc Neil A., Jedrzejczyk A., Armaganijan L. A systematic review of
depression and anxiety in patients with atrial fibrillation: the mind-heart link. Cardiovasc. Psychiatry Neurolog.
2013; 2013: 159850. doi: 10.1155/2013/159850.
8. Hindricks G., Potpara R., Dagres N., Arbelo E., Bax J.J., Blomström-Lundqvist C., Borianin G., Castella M.,
Dan G.A., Dilaveris P.E., Fauchier L., Filippatos G., Kalman J.M., La Meir M., Lane D.A., Lebeau J.P. 2020 ES
Guidelines for the diagnosis and management of atrial fibrillation developed in collaboration with the
European Association for Cardio-Thoracic Surgery (EACTS). Eur. Heart. J. 2021; 42 (5): 373-498. doi:
10.1093/eurheartj/ehaa612.
9. Sang C.H., Chen K., Pang X.F., Dong J.Z., Du X., Ma H., Liu J.H., Ma C.S., Sun Y.X. Depression, anxiety and
quality of life after catheter ablation in patients with paroxysmal atrial fibrillation. Clin. Cardiol. 2013; 36 (1):
40-45. doi: 10.1002/clc.22039.
10. Jain S.K., Novak P.G., Sangrigoli R., Champagne J., Dubuc M., Adler S.W., Svinarich J.T., Essebag V., Martien
M., Anderson C., John R.M., Mansour M., Knight B.P. Sustained quality-of-life improvement post-cryoballoon
ablation in patients with paroxysmal atrial fibrillation: Results from the STOP-AF Post-Approval Study. Heart
Rhythm. 2020; 17 (3): 485-491. doi: 10.1016/j.hrthm.2019.10.014.
11. Wedegärtner S.M., Schwantke I., Kindermann I., Karbach J. Predictors of heart-focused anxiety in patients
with stable heart failure. J. Affect. Disord. 2020; 276: 380-387. doi: 10.1016/j.jad.2020.06.065.
12. Kindermann I., Wedegärtner S.M., Bernhard B., Ukena J., Lenski D., Karbach J., Schwantke I., Ukena C.,
Böhm M. Changes in quality of life, depression, general anxiety, and heart-focused anxiety after
defibrillator implantation. ESC Heart Failure. 2021; 8: 2502-2512. doi: 10.1002/ehf2.13416.
13. Hohls J.K., Beer K., Arolt V., Haverkamp W., Kuhlmann S.L., Martus P., Waltenberger J., Rieckmann N.,
Müller-Nordhorn J., Ströhle A. Association between heart-focused anxiety, depressive symptoms, health
behaviors and healthcare utilization in patients with coronary heart disease. J Psychosom Res. 2020; 11; 131:
109958. doi: 10.1016/j.jpsychores.2020.109958. Epub ahead of print.
14. Eifert G.H., Zvolensky M.J., Lejuez C.W. Heart-Focused Anxiety and Chest Pain: A Conceptual and Clinical
Review. Clinical Psychology: Science and Practice. 2000; 7: 403-417. doi: 10.1093/clipsy.7.4.403.
15. Fischer D., Kindermann I., Karbach J., Herzberg P.Y., Ukena C., Barth C., Lenski M., Mahfoud F., Einsle F.,
Dannemann S., Böhm M., Köllner V. Heart-focused anxiety in general population. Clin. Res. Cardiol. 2012; 101
(2): 109-16. doi: 10.1007/s00392-011-0371-7.
16. Carmen Terol-Cantero M., Cabrera-Perona V., Martín-Aragón M. Hospital Anxiety and Depression Scale
(HADS) review in Spanish samples. Annals of Psychology. 2015; 31 (2): 494-503. doi:
10.6018/analesps.31.2.172701.
17. Bjelland I., Dahl A.A., Haug T.T., Neckelmann D. The validity of the Hospital Anxiety and Depression Scale.
An updated literature review. J. Psychosom. Res. 2002; 52 (2): 69-77. doi: 10.1016/S0022-3999(01)00296-3.
18. Ware J.Jr., Kosinski M., Keller S.D. A 12-Item Short-From Health Survey: construction of scales and
preliminary tests of reliability and validity. Med. Care. 1996; 34 (3): 220-33. doi: 10.1097/00005650-199603000-
00003.
19. Gandek B., Ware J.E., Aaronson N.K., Apolone G., Bjorner J.B, Brazier J.E., Bullinger M., Kaasa S., Leplege A.,
Prieto L., Sullivan M. Cross-validation of item selection and scoring for the SF-12 Health Survey in nine
countries: results from the IQOLA Project. Internatinal Quality of Life Assessment. J. Clin. Epidemiol. 1998; 51
(11): 1171-8. doi: 10.1016/s0895-4356(98)00109-7.
20. Bullinger M., Kirchberger I., Ware J. Der deutsche SF-36 Health Survey Übersetzung und psychometrische
Testung eines krankheitsübergreifenden Instruments zur Erfassung der gesundheitsbezogenen
Lebensqualität. Z. f. Gesundheitswiss. 1995; 3 (21). doi: 10.1007/BF02959944.
21. Zhuo C., Ji F., Lin X., Jiang D., Wang L., Tian H., Xu Y., Liu S., Chen C. Depression and recurrence of atrial
fibrillation after catheter ablation: a meta-analysis of cohort studies. J. Affect. Disord. 2020; 271: 27-32. doi:
10.1016/j.jad.2020.03.118.
22. Koleck T.A., Mitha S.A., Biviano A., Caceres B.A., Corwin E.J., Goldenthal I., Creber R.M., Turchioe M.R.,
Hickey K.T., Bakken S. Exploring Depressive Symptoms and Anxiety Among Patients With Atrial Fibrillation
and/or Flutter at the Time of Cardioversion and Ablation. J. Cardiovasc. Nurs. 2021; 36 (5): 470-481. doi:
10.1097/JCN.0000000000000723.
23. Walfridsson H., Walfridsson U., Nielsen J.C., Johannessen A., Raatikainen P., Janzon M., Levin L.A., Aronsson
M., Hindricks G., Kongstad O., Pehrson S., Englund A., Hartikainen J., Mortensen L.S., Hansen P.S.
Radiofrequency ablation as initial therapy in paroxysmal atrial fibrillation: results on health-related quality of
life and symptom burden. The MANTRA-PAF trial. Europace. 2015; 17 (2): 215-21. doi:
10.1093/europace/euu342.
24. Poole J., Bahnson T., Monahan K., Johnson G., Rostami H., Silverstein A., Al-Khalidi H., Rosenberg Y., Mark
d., Lee K., Packer D., Akoum N., Aoukar P., Birgersdotter-Green U., Blatt J., Cha Y., Chung M., Gleva M.,
Glotzer T., Henrickson C., Kron J., Kuriachan V., Mulpuru S., Noseworthy P., Patton K., Prutkin J., Ranjan R.,
Rho R., Russo A., Stecker E., Tzou W., Serdoz l., Wilson M. Recurrence of Atrial Fibrillation After Catheter
Ablation or Antiarrhythmic Drug Therapy in the CABANA Trial. JACC. 2020; 75 (25): 3119-21.
Doi: 10.1016/j.jacc.2020.04.065.
25. Björkenheim A., Brandes A., Magnuson A., Chemnitz A., Edvardsson N., Poçi D. Patient-Reported Outcomes
in Relation to Continuously Monitored Rhythm Before and During 2 Years After Atrial Fibrillation Ablation
Using a Disease-Specific and a Generic Instrument. J. Am. Heart Assoc. 2018; 7 (5): e008362. Doi:
10.1161/JAHA.117.008362.
